# Induction Therapies Determine the Distribution of Perforin and Granzyme B Transcripts in Kidney Transplant Recipients

**DOI:** 10.3390/biomedicines12061258

**Published:** 2024-06-05

**Authors:** Dino Pipic, Marianne Rasmussen, Qais W. Saleh, Martin Tepel

**Affiliations:** 1Institute of Molecular Medicine, Clinical Institute, University of Southern Denmark, 5000 Odense, Denmark; 2Department of Nephrology, Odense University Hospital, 5000 Odense, Denmark

**Keywords:** Perforin, Granzyme B, kidney transplant recipients, induction therapies, thymoglobulin, defense against pathogens

## Abstract

Peripheral blood mononuclear cells contain secretory granules with Perforin and Granzyme B for defense against pathogens. The objective of the present study was to compare the effects of immunosuppressive induction therapies on Perforin and Granzyme B transcripts in kidney transplant recipients. Transcripts were determined in 408 incident kidney transplant recipients eight days posttransplant using quantitative real-time PCR. Compared to 90 healthy subjects, the median Perforin transcripts were lower in kidney transplant recipients with blood-group ABO-incompatible donors (N = 52), compatible living donors (N = 130), and deceased donors (N = 226) (25.7%; IQR, 6.5% to 46.0%; 31.5%; IQR, 10.9% to 57.7%; and 35.6%; IQR, 20.6% to 60.2%; respectively; *p* = 0.015 by the Kruskal–Wallis test). Kidney transplant recipients who were treated with thymoglobulin (N = 64) had significantly lower Perforin as well as Granzyme B compared to all other induction therapies (N = 344) (each *p* < 0.001). Receiver operator characteristics analysis showed that both Perforin (area under curve, 0.919) and Granzyme B (area under curve, 0.915) indicated thyroglobulin-containing induction therapies. Regression analysis showed that both reduction in plasma creatinine and human leukocyte antigen mismatches were positively associated with elevated Perforin/Granzyme B transcript ratio posttransplant. We conclude clinical parameters and therapies affect Perforin and Granzyme B transcripts posttransplant.

## 1. Introduction

Human peripheral blood mononuclear cells (PBMCs) constitutively express the pore-forming cytolytic protein, Perforin, and the serine protease, Granzyme B [1]. PBMCs as well as cytotoxic T lymphocytes, i.e., CD8+ cytotoxic T lymphocytes and natural killer cells, help defend against pathogens [2]. Single-cell RNA sequencing and single-cell T-cell receptor sequencing identified a causal role for activated cytotoxic T lymphocytes, for example, in kidneys of patients with progressive forms of immune-mediated kidney disease, i.e., antineutrophilic cytoplasmic antibody (ANCA)-associated vasculitis [3]. PBMCs and cytotoxic T lymphocyte recognize their target cell; then, Perforin and Granzyme B are released from their secretory granules, and finally, they cause granule-dependent target cell death. Perforin forms large transmembrane pores, then Granzyme B diffuses into the target cell and initiates apoptosis [4,5,6]. Studies in animal models supported these mechanisms, since the depletion of cytotoxic lymphocytes reduced apoptosis in mice [7]. Furthermore, mice with Granzyme B-deficient CD8+ cytotoxic lymphocytes showed reduced apoptosis [7]. Beyond kidney diseases, elevated Granzyme B has also been shown to enhance cardiovascular diseases [6]. Increased Granzyme B plasma levels have been associated with increased mortality in patients with acute myocardial infarction [7].

There are several induction therapies in kidney transplant recipients, including basiliximab, rituximab, prednisolone, and thymoglobulin. Induction therapies for kidney transplant recipients are chosen on the basis of clinical experiences and guidelines. It is suspected that different induction therapies may show different side effects including infections. Since Perforin and Granzyme B are major players in the defense against pathogens, induction therapies may affect infections by interfering with Perforin and Granzyme B. Currently, it is unknown whether different immunosuppressive induction therapies, including basiliximab, rituximab, prednisolone, and thymoglobulin, may cause separate effects on Perforin and Granzyme B transcripts in peripheral blood mononuclear cells. The objective of the present study was to compare the effects of immunosuppressive induction therapies on Perforin and Granzyme B transcripts posttransplant in mononuclear cells from kidney transplant recipients. We investigated transcripts in mononuclear cells from a large cohort of kidney transplant recipients [8,9,10,11,12]. We investigated the effects of different induction therapies. According to the literature, the relative reduction in creatinine was used as an outcome measure [13]. We showed for the first time that immunosuppressive induction therapies as well as kidney function were associated with Perforin and Granzyme B transcript levels posttransplant.

## 2. Materials and Methods

### 2.1. Study Design and Patient Population

Transplant recipients were recruited for the ongoing “Molecular Monitoring after Kidney Transplantation (MoMoTx)” study, ClinicalTrials.gov Identifier is NCT01515605. In the study, healthy control subjects were also recruited among transplant donors. The study was conducted following the ethical guidelines outlined in the Declaration of Helsinki and Istanbul. The local ethics committee, “Den Videnskabsetiske Komite for Region Syddanmark”, approved the study protocol (identification code: 20100098). Written informed consent was obtained from patients before enrollment in the study. Inclusion criteria were incident kidney transplantation, age above 18 years, and informed consent. Details from the “MoMoTx” study had been published previously [8,9,10,11,12]. Clinical characteristics, information about organ procurement, and laboratory data were obtained from medical records. The clinicians followed local protocols to determine the immunosuppressive therapy and concomitant therapies. The relative decrease in plasma creatinine on the first postoperative day was calculated as the difference between pretransplant and first-day posttransplant plasma creatinine divided by pretransplant plasma creatinine [13].

### 2.2. Isolation of Peripheral Blood Mononuclear Cells

Blood samples were taken on the eighth postoperative day after kidney transplantation. Peripheral blood mononuclear cells (PBMCs) were isolated from heparinized whole blood according to the method previously described by our group [8,9,10,11,12]. This process was carried out by density gradient centrifugation using Histopaque (Sigma Aldrich; Søborg, Denmark; density: 1.077 g/mL). We centrifuged heparinized blood for 4 min at 1620× *g* and removed the supernatant. The blood was diluted with 1.5 mL of Hanks’ Balanced Salt solution (HBSS), layered on 1.8 mL of Histopaque, and centrifuged for 15 min at 952× *g* at room temperature. PBMCs were carefully harvested from the interphase and washed in phosphate-buffered saline by centrifugation for 3 min at 8000× *g*. The supernatant was decanted, and the precipitate was suspended in 400 μL TRI reagent (TRIzol, Sigma Aldrich, Søborg, Denmark) and stored at minus 80 °C.

### 2.3. Purification of Total RNA and Synthesis of Complementary DNA

The purification of RNA and synthesis of cDNA was performed as described [8,9]. Briefly, extraction and purification of total RNA were carried out by using an RNeasy mini kit including an RNase-free DNase set (Qiagen, Hilden, Germany) according to the manufacturer’s protocol. The concentrations and purity of total RNA were assessed in duplicate measurements using a UV-visible spectrophotometer (Implen nanophotometer). We measured the absorbance at 260 nm and 280 nm, and the ratio (A260/A280) was used to assess RNA purity. Samples that had a ratio between 1.8 and 2.1 indicated a highly purified RNA and were included for further processing.

The extracted RNA was reverse transcribed into cDNA using a QuantiTect Reverse Transcription Kit (Qiagen, Hilden, Germany). An amount of 300 ng of total RNA was incubated in gDNA wipeout buffer at 42 °C for 4 min to eliminate genomic DNA effectively. To initiate reverse transcription, total RNA was mixed with Reverse-transcription Master Mix (Quantiscript Reverse Transcriptase, Quantiscript RT buffer, and RT Primer Mix) followed by incubation at 60 °C for 60 min. The reaction was inactivated at 95° for 5 min, and the cDNA was stored at minus 20 °C until further analysis.

### 2.4. Primers and Quantitative Reverse Transcriptase Real-Time Polymerase Chain Reaction (qRT-PCR)

Quantifications of Granzyme B and Perforin were performed using quantitative reverse transcriptase real-time polymerase chain reaction (qRT-PCR). Gene-specific oligonucleotide primers were designed using the Primer-BLAST Software tool (National Center for Biotechnology Information; Bethesda, MD, USA; https://www.ncbi.nlm.nih.gov/tools/primer-blast/, accessed on 22 March 2023) and obtained from Merck, Darmstadt, Germany. The target base pair (bp) PCR products for Granzyme B, Perforin, and β-actin transcripts were 100 bp, 173 bp, and 234 bp, respectively. Transcripts were measured using primers given in Appendix A.

According to the manufacturer’s instructions, the qRT-PCR measurements were conducted using the LightCycler 96 Instrument (Roche, Copenhagen, Denmark). The cycling conditions were as follows: 1 cycle of preincubation at 95 °C for 10 min followed by 50 cycles of a 3-step amplification: denaturation of the template at 95 °C for 10 s, annealing of primers to template at 60 °C for 10 s, and extension of primers at 72 °C for 10 s. The amplification runs were performed using the FastStart Essential DNA Green Master Mix (Roche, Copenhagen, Denmark, LOT: 64697500). Each reaction well had a final volume of 20 μL, which contained 2 μL of cDNA, 4 μL of nuclease-free water, 2 μL of each primer, and 10 μL of FastStart Essential DNA Green Master Mix.

Amplification plots and melting peaks were visually inspected, and quantification cycle (Cq) values were evaluated using LightCycler 96 Software 1.1 (Roche, Copenhagen, Denmark). The Cq values were derived from amplification plots where the fluorescence was higher than the threshold level. We included in-house cDNA control and nuclease-free water control in each PCR plate run. All reactions were run in duplicate.

The target gene expressions were determined relative to the β-actin and normalized ratios of transcript expression were calculated according to the following equation:Normalized ratio = ET^CqR-CqT^
with ET being the efficiency of target amplification and CqT and CqR being the quantification cycle at target/reference detection.

To calculate the relative fold gene expression of samples, Cq values were also normalized to values from healthy control subjects according to the delta–delta quantification cycle (ddCq) method (2^−∆∆Cq^ method).

### 2.5. Determination of Primer Efficiency

Primer efficiencies for Perforin, Granzyme B, and β-actin were generated from a standard curve method by serial dilutions of selected dsDNA templates as shown in Appendix A. Primer efficiency was calculated using the following equation: E = slope/10. The purification of dsDNA from PCR products was carried out on a 100 μL in-house calibrator sample by using a QIAquick PCR Purification Kit (Qiagen, Hilden, Germany, catalog nr: 28014) according to the manufacturer’s protocol. The concentration and purity of dsDNA were evaluated and the acceptable A260/A280 ratio was in the range between 1.8 and 2.1. The concentration of dsDNA (stock solution) was measured, and then serial dilutions were performed (Appendix A).

### 2.6. Electrophoresis of PCR Products

Electrophoresis through 2% agarose gel (Thermo Scientific GeneRuler; Roskilde, Denmark) was used to size-fraction and visually control the band size of PCR products (Appendix A). A 2% agarose gel was made by mixing 30 mL of TAE buffer with 0.6 g of agarose powder. The mixture was microwaved for 1–3 min until the agarose was completely dissolved. We then added 3 μL of GelRed 10,000× stock reagent into the solution at 1:100,000 dilution. The agarose was slowly poured into a gel tray with the well comb in place, and the gel was completely solidified at room temperature within 30 min. The chamber was then filled with 1× TAE buffer until the gel was covered. A Gene Ruler 50 bp DNA ladder (Thermo Scientific, catalog nr: SM0371; Roskilde, Denmark) was loaded into three different wells of the gel to assess the size of PCR products visually. An amount of 4 μL of sample buffer together with 20 μL of PCR products of Perforin, Granzyme B, and β-actin were mixed with 4 μL Gel Loading Dye, Blue (6×) (New England BioLabs; Copenhagen, Denmark) and loaded into separate, labeled wells. We performed electrophoresis at 100 V for approximately 1 h, and ultraviolet (UV) light was used to visualize the band size of PCR products.

### 2.7. Statistical Analysis

For descriptive analysis, continuous data are presented as the median and interquartile range (IQR). Continuous data were compared using the non-parametric Mann–Whitney test or Kruskal–Wallis test as appropriate. Categorical data are reported as frequency counts with numbers (percent). Categorical data were compared using Fisher’s exact or chi-squared test as appropriate.

We performed receiver operator characteristic (ROC) curve analysis to detect the accuracy of transcripts of Perforin or Granzyme B to predict induction therapy containing thymoglobulin versus other induction therapies. The cutoff value was determined using the Youden index.

Univariable regression analyses for predictors of Perforin/Granzyme B transcript ratio in incident kidney transplant recipients were performed, which included recipient age, recipient gender, systolic and diastolic blood pressure, the relative reduction in plasma creatinine on the first postoperative day, i.e., an early indicator of allograft function after transplant, and number of HLA mismatches. Data were analyzed using GraphPad Prism software (version 6.0, GraphPad Software, La Jolla, CA, USA) and SPSS (version 26.0.1.0; IBM Corp., IBM SPSS Statistics for Windows, Armonk, NY, USA). All statistical tests were two-sided. Two-sided *p*-values less than 0.05 were considered to indicate statistical significance.

## 3. Results

### 3.1. Perforin and Granzyme B Transcripts in Recipients after Kidney Transplantation

We determined Perforin and Granzyme B transcripts in peripheral blood mononuclear leukocytes in 408 incident kidney transplant recipients eight days after incident kidney transplantation. The flow chart is depicted in Figure 1. A total of 52 recipients had blood group AB0-incompatible living donor allografts (ABOi), 130 recipients had ABO-compatible living donor allografts (LD), and 226 recipients had deceased donor allografts (DD). The clinical and biochemical characteristics of kidney transplant recipients are given in Table 1 and Table 2. Induction therapies in kidney transplant recipients are shown in Table 3.

Median Perforin transcripts were 0.005647 (IQR, 0.001469 to 0.010120) in blood group AB0i, 0.00688 (IQR, 0.00244 to 0.01263) in LD, and 0.00790 (IQR, 0.00454 to 0.01322) in DD, respectively (*p* = 0.015 by Kruskal–Wallis test; Figure 2A). Dunn’s multiple comparisons test revealed that Perforin transcripts were significantly lower in ABOi compared to DD (*p* < 0.05).

Median Granzyme B transcripts were 0.00214 (IQR, 0.00060 to 0.00402) in blood group AB0i, 0.00414 (IQR, 0.00141 to 0.00900) in LD, and 0.00610 (IQR, 0.00305 to 0.01047) in DD (*p* < 0.0001 by Kruskal–Wallis test; Figure 2B). Dunn’s multiple comparisons test revealed that Granzyme B transcripts were significantly lower in ABOi compared to LD (*p* < 0.01), in ABOi compared to DD (*p* < 0.001), and in LD compared to DD (*p* < 0.01).

In 90 healthy control subjects, the median Perforin transcripts were 0.021134 (IQR, 0.01426 to 0.03248), and the median Granzyme B transcripts were 0.01405 (IQR, 0.00851 to 0.02130).

Figure 3 shows histograms of the relative frequencies and the cumulative distributions of the Perforin/Granzyme B ratio in 408 kidney transplant recipients (Figure 3A) and 90 healthy control subjects (Figure 3B). In kidney transplant recipients, the median Perforin/Granzyme B ratio was 1.47 (IQR 0.88 to 2.36). In healthy control subjects, the median Perforin/Granzyme B ratio was 1.75 (IQR 1.14 to 2.29; *p* = 0.12 by Mann–Whitney test).

When quantifying transcripts according to the ddCq method, we observed that ABOi had a median of 25.7% (IQR, 6.5% to 46.0%) Perforin transcripts of healthy subjects, LD had 31.5% (IQR, 10.9% to 57.7%) Perforin transcripts of healthy subjects, and DD had 35.6% (IQR, 20.6% to 60.2%) Perforin transcripts of healthy subjects, respectively (*p* = 0.015 by the Kruskal–Wallis test; Figure 4A).

When quantifying transcripts according to the ddCq method, we observed that ABOi had a median of 14.5% (IQR, 3.5% to 29.6%) Granzyme B transcripts of healthy subjects, LD had 29.0% (IQR, 9.2% to 64.4%) Granzyme B transcripts of healthy subjects, and DD had 40.7% (IQR, 21.4% to 73.0%) Granzyme B transcripts of healthy subjects, respectively (*p* < 0.0001 by the Kruskal–Wallis test; Figure 4B).

### 3.2. Effect of Induction Therapies on Perforin and Granzyme B Transcripts after Transplantation

Now we quantified Perforin and Granzyme B transcripts in kidney transplant recipients according to different induction therapies. A total of 328 kidney transplant recipients (80%) were treated with basiliximab (Ba), 89 recipients (22%) with rituximab (Ri), 121 recipients (30%) with methylprednisolone (Pre), and 64 recipients (16%) with thymoglobulin (TGL), respectively. The total number exceeds 100% because recipients obtained combinations of more than one immunosuppressive agent. In 52 kidney transplant recipients with ABOi, 1 was treated with Ba (2%), 1 with BaPre (2%), 26 with BaRiPre (50%), 15 with RiPre (29%), and 9 with RiPreTGL (17%), respectively. In 130 kidney transplant recipients with LD, 97 were treated with Ba (75%), 3 with BaPre (2%), 4 with BaRiPre (3%), 16 with RiPreTGL (12%), 1 with TGL (1%), and 9 with PreTGL (7%), respectively. In 226 kidney transplant recipients with DD, 186 were treated with Ba (82%), 10 with BaPre (4%), 1 with BaRiRe (0%), 18 with RiPreTGL (8%), 2 with TGL (1%), and 9 with PreTGL (4%), respectively.

As shown in Figure 5, in kidney transplant recipients with ABOi, median Perforin transcripts were significantly lower in recipients who obtained RiPreTGL (median, 0.0002190; IQR, 0.00008 to 0.00064; N = 9) compared to BaRiPre (median 0.00694; IQR, 0.00275 to 0.01236; N = 26; *p* < 0.001) and RiPre (0.00599; IQR, 0.00348 to 0.01214; N = 15; *p* < 0.01 by the Kruskal–Wallis test and Dunn’s multiple comparisons test).

In LD, Perforin transcripts were significantly lower in recipients who obtained PreTGL compared to Ba (*p* < 0.05), RiPreTGL compared to Ba (*p* < 0.001), and RiPreTGL compared to BaRiTGL (*p* < 0.05 by the Kruskal–Wallis test and Dunn’s multiple comparisons test).

In DD, Perforin transcripts were significantly lower in recipients who obtained RiPreTGL compared to Ba (*p* < 0.001); RiPreTGL compared to BaPre (*p* < 0.001), PreTGL compared to Ba (*p* < 0.001), and PreTGL compared to BaPre (*p* < 0.001 by the Kruskal–Wallis test and Dunn’s multiple comparisons test).

As shown in Figure 6, in kidney transplant recipients with ABOi, median Granzyme B transcripts were significantly lower in recipients who obtained RiPreTGL (median; 0.00027, IQR, 0.00014 to 0.00040; N = 9) compared to BaRiPre (median 0.00299; IQR, 0.00135 to 0.00502; N = 26; *p* < 0.001) and RiPre (median, 0.00182; IQR, 0.00090 to 0.00563; N = 15; N < 0.01 by the Kruskal–Wallis test and Dunn’s multiple comparisons test).

In LD, Granzyme B transcripts were significantly lower in recipients who obtained PreTGL compared to Ba (*p* < 0.05), RiPreTGL compared to Ba (*p* < 0.001), and RiPreTGL compared to BaRiTGL (*p* < 0.01 by the Kruskal–Wallis test and Dunn’s multiple comparisons test).

In DD, Granzyme transcripts were significantly lower in recipients who obtained RiPreTGL compared to Ba (*p* < 0.001), RiPreTGL compared to BaPre (*p* < 0.001), PreTGL compared to Ba (*p* < 0.001), and PreTGL compared to BaPre (*p* < 0.01 by the Kruskal–Wallis test and Dunn’s multiple comparisons test).

Kidney transplant recipients who were treated with thymoglobulin had significantly lower Perforin levels compared to all other induction therapies (median, 0.00046; IQR, 0.00014 to 000187; N = 64; vs. median, 0.00865; IQR, 0.00497 to 0.01355; N = 344; *p* < 0.001 by the Mann–Whitney test). Kidney transplant recipients who were treated with thymoglobulin also had significantly lower Granzyme B levels compared to all other induction therapies (median, 0.00048; IQR, 0.00017 to 000136; N = 64; vs. median, 0.00590; IQR, 0.00320 to 0.01045; N = 344; *p* < 0.001 by the Mann–Whitney test). Receiver operator characteristic (ROC) curve analysis showed that both Perforin (Figure 7A) and Granzyme B (Figure 7B) indicated thymoglobulin-containing induction therapies. The cutoff value was determined using the Youden index. The Perforin level of 0.00889 showed a sensitivity of 0.488 (95% CI, 0.434 to 0.543) and a specificity of 0.969 (95% CI, 0.892 to 0.996; likelihood ratio, 15.63). The Granzyme B level of 0.00770 showed a sensitivity of 0.284 (95% CI, 0.332 to 0.437) and a specificity of 0.984 (95% CI, 0.916 to 0.999; likelihood ratio, 24.56). We performed sensitivity analyses for 226 kidney transplant recipients who received deceased donor allografts. A total of 29 kidney transplant recipients with deceased donor allografts had thymoglobulin-containing induction therapies, whereas 197 had other induction therapies. For Perforin, the ROC curves showed an area under curve of 0.920 (95% CI, 0.858 to 0.982; *p* < 0.001), and for Granzyme B, the ROC curves showed an area under curve of 0.925 (95% CI, 0.868 to 0.982; *p* < 0.001).

### 3.3. Evaluation of Predictors for Perforin/Granzyme B Transcript Ratio after Transplantation

Next, we investigated the Perforin/Granzyme B transcript ratio in 226 recipients after kidney transplantation with deceased donor allograft. The ratio of Perforin/Granzyme B transcripts was significantly higher than the theoretical value of one (median, 1.3114; IQR, 0.8445 to 1.9944; *p* < 0.001 by Wilcoxon Signed Rank Test), indicating independent regulations of these transcripts after kidney transplant. We used univariable regression analysis to evaluate clinical and biochemical variables that may predict Perforin/Granzyme B transcript ratio posttransplant (Table 4). A larger relative decrease in plasma creatinine on the first postoperative day as well as higher human leukocyte antigen mismatches were positively associated with a higher Perforin/Granzyme B transcript ratio. This finding may indicate that an increased activation of the innate immune system mainly activates pore-forming defense mechanisms.

## 4. Discussion

The present study in 408 incident kidney transplant recipients indicates that induction therapies affect Perforin and Granzyme B transcripts in mononuclear cells. Both Perforin and Granzyme B transcripts were significantly lower in kidney transplant recipients compared to healthy controls. Recipients with blood group AB0-incompatible donors had lower Perforin as well as Granzyme B transcripts compared to compatible living donors and deceased donors. We showed that a thymoglobulin-containing induction therapy caused significantly lower Perforin as well as Granzyme B transcripts compared to all other induction therapies.

### 4.1. Determination of the Extent of Immunosuppression after Kidney Transplantation

We showed that transcripts of Perforin, as well as Granzyme B, were significantly lower in 408 kidney transplant recipients compared to 90 healthy subjects. Cytotoxic T lymphocytes, i.e., CD8+ cytotoxic T lymphocytes and natural killer cells, are the major sources of Perforin and Granzyme B. Several induction therapies including basiliximab, rituximab, prednisolone, and thymoglobulin are commonly used because they showed superior results in reducing renal allograft rejection and/or allograft failure compared to maintenance therapy alone [14]. It is well known that immunosuppressive induction therapies in kidney transplant recipients affect cytotoxic T lymphocytes. Now, we showed that the immunosuppressive effect is not uniform for all kidney transplant recipients. We observed that the median transcript levels in 52 recipients with blood group AB0-incompatible donors were significantly lower in blood group AB0i compared to 130 recipients with LD and 226 recipients with DD allografts. Median Perforin transcript levels were 25.7%, 31.5%, and 35.6% of healthy subjects in kidney transplant recipients with blood group AB0i, LD, and DD allografts, respectively. Median Granzyme B transcript levels were 14.5%, 29.0%, and 40.7% of healthy subjects in kidney transplant recipients with blood group AB0i, LD, and DD allografts, respectively. Currently, there is no assay available to accurately determine the extent of immunosuppression after kidney transplantation [15]. Determination of Perforin and Granzyme B transcripts in peripheral blood mononuclear cells may help to determine the extent of immunosuppression.

### 4.2. Mechanisms That Regulate the Expression of Perforin and Granzyme B in Peripheral Blood Mononuclear Cells

Cytokines stimulate the production, trafficking, and storage of large amounts of Perforin and Granzymes in secretory granules [2]. Under pathophysiological conditions, approximately 8 days after acute viral infection, Perforin mRNA is upregulated in effector CD8+ T lymphocytes [16]. Perforin expression is regulated transcriptionally and correlates tightly with the development of cells that can exhibit cytotoxic activity [16]. MicroRNA, e.g., microRNA-150, affects the expression of Perforin [17]. Perforin protein is predominantly expressed in effector CD8+ T lymphocytes, which are a minority of all CD8+ T lymphocytes in the peripheral blood, but which are largely expanded during acute viral infection [18].

Thymoglobulin is a polyclonal antibody that causes depletion of T-cells in the peripheral blood mononuclear cells and probably also in lymphoid tissues by binding to the T-cell surface antigen [19]. However, thymoglobulin may also bind B-lymphocytes and dendritic cells [19]. Thymoglobulin causes complement-dependent lysis of T-cells, apoptosis, and subsequent phagocytosis by macrophages [20]. In line with these findings, the present study confirmed that induction therapies containing thymoglobulin caused a reduction in both Perforin and Granzyme B transcripts after transplantation. Kidney transplant recipients who received thymoglobulin-containing induction therapy had significantly lower Perforin levels as well as Granzyme B levels compared to all other induction therapies.

Downregulation of Perforin and Granzyme B in human PBMCs has been reported earlier [21,22]. First, the reduced expression of Perforin and Granzyme B in cultured human PBMCs after the addition of Interleukin-6 was ameliorated by the Interleukin-6 receptor blocker, tocilizumab. Furthermore, Interleukin-6 transgenic mice had reduced cell cytotoxicity and showed lower expression of Perforin and Granzyme B compared to wild-type mice [21,22]. Second, Ajith et al. investigated human leukocyte antigen (HLA)-G, a non-classic HLA class Ib molecule that has been described to cause maternal tolerance to fetal tissue. Soluble HLA-G dimer levels were higher in 90 patients with a functioning renal allograft compared with 40 patients with allograft rejection. Flow cytometry showed an increased expression of HLA-G on CD8+ T cells from patients with functioning renal allografts. Importantly, HLA-G dimers downregulated the expression of Granzyme B in human PBMCs, which may prevent allograft rejection and prolong graft survival [23]. Third, the organophosphorus compound, dimethyl 2,2-dichlorovinyl phosphate (DDVP), reduces the expression of Perforin and Granzyme B in NK cells [24].

Krepsova et al. investigated the effect of induction therapies on the lymphocyte subpopulations using FACS and the expression of biomarkers, including Perforin and Granzyme B in whole blood samples from 6o kidney transplant recipients. They reported that recipients who obtained thymoglobulin induction therapy showed a reduction in NK cells as well as Perforin and Granzyme B transcripts [25]. Simon et al. reported a significant reduction in Perforin and Granzyme B expression after administration of thymoglobulin in 11 kidney transplant recipients [26]. We observed that induction therapies affect Perforin and Granzyme B transcripts in mononuclear cells from kidney transplant recipients.

Limitation: The current study lacks an infection analysis, and such an analysis should be performed in future investigations.

We observed that several induction therapies have distinct effects on immune cells. Future studies should determine whether these differences may have an impact on both kidney function as well as complications after kidney transplant, for example, specific infections.

## Figures and Tables

**Figure 1 biomedicines-12-01258-f001:**
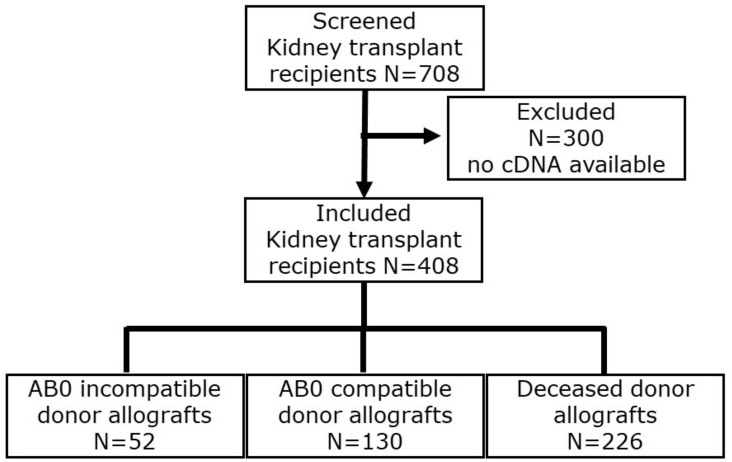
Flow chart of incident kidney transplant recipients.

**Figure 2 biomedicines-12-01258-f002:**
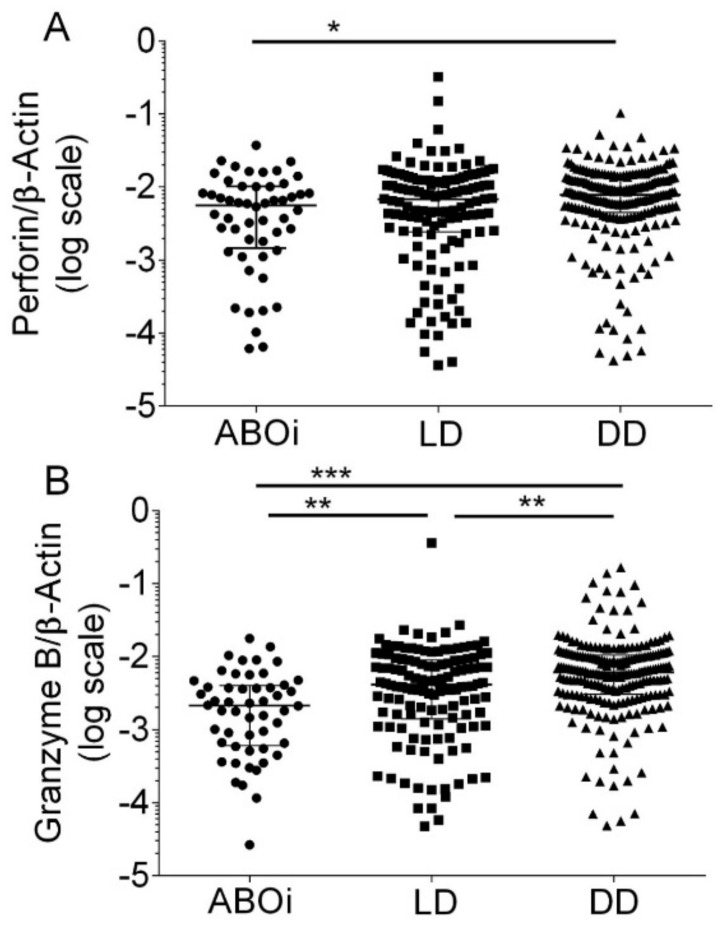
Scatter diagram of Perforin transcripts (**A**) and Granzyme B transcripts (**B**) in peripheral blood mononuclear leukocytes from 408 recipients 8 days after incident kidney transplantation. A total of 52 recipients had blood group AB0-incompatible living donor allografts (ABOi), 130 recipients had ABO-compatible living donor allografts (LD), and 226 recipients had deceased donor allografts (DD). Median values and interquartile range (IQR) are indicated. Differences between groups were analyzed with the Kruskal–Wallis test and Dunn’s multiple comparisons test, * *p* < 0.05; ** *p* < 0.01; *** *p* < 0.001).

**Figure 3 biomedicines-12-01258-f003:**
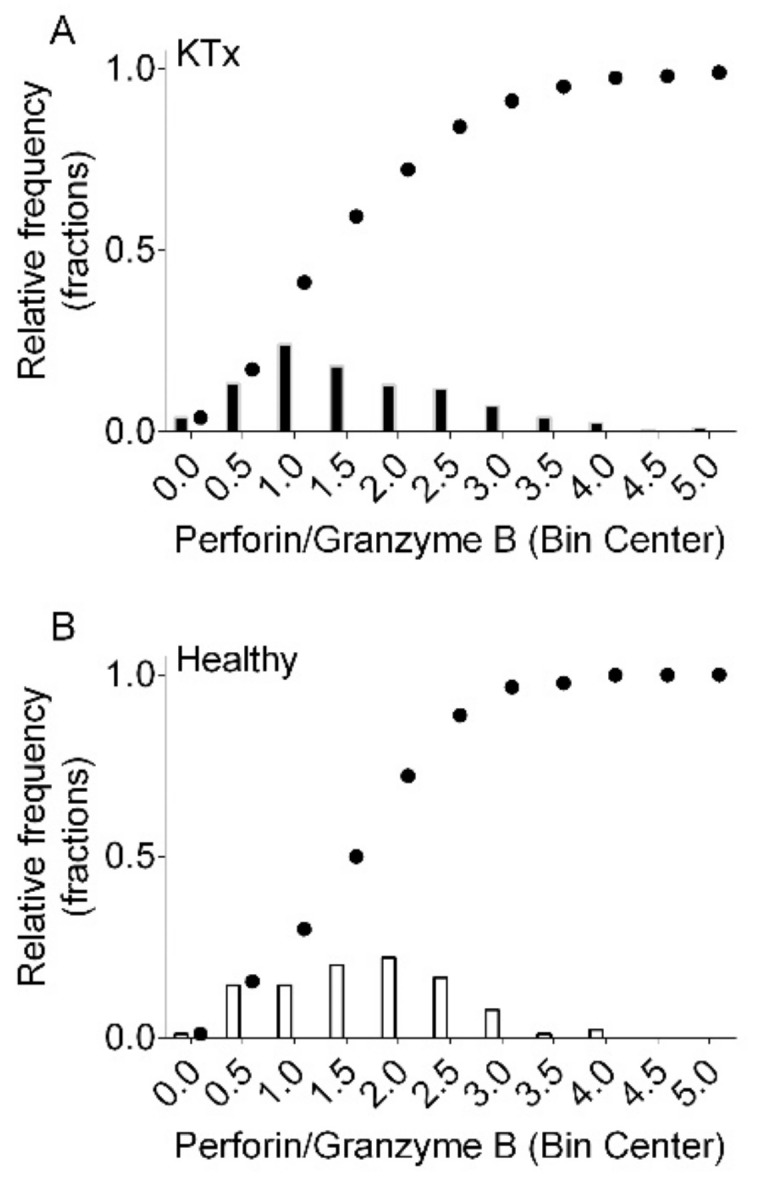
Histograms of the relative frequencies and cumulative distributions of the Perforin/Granzyme B ratio in 408 kidney transplant recipients (KTx (**A**) and 90 healthy control subjects (Healthy (**B**)).

**Figure 4 biomedicines-12-01258-f004:**
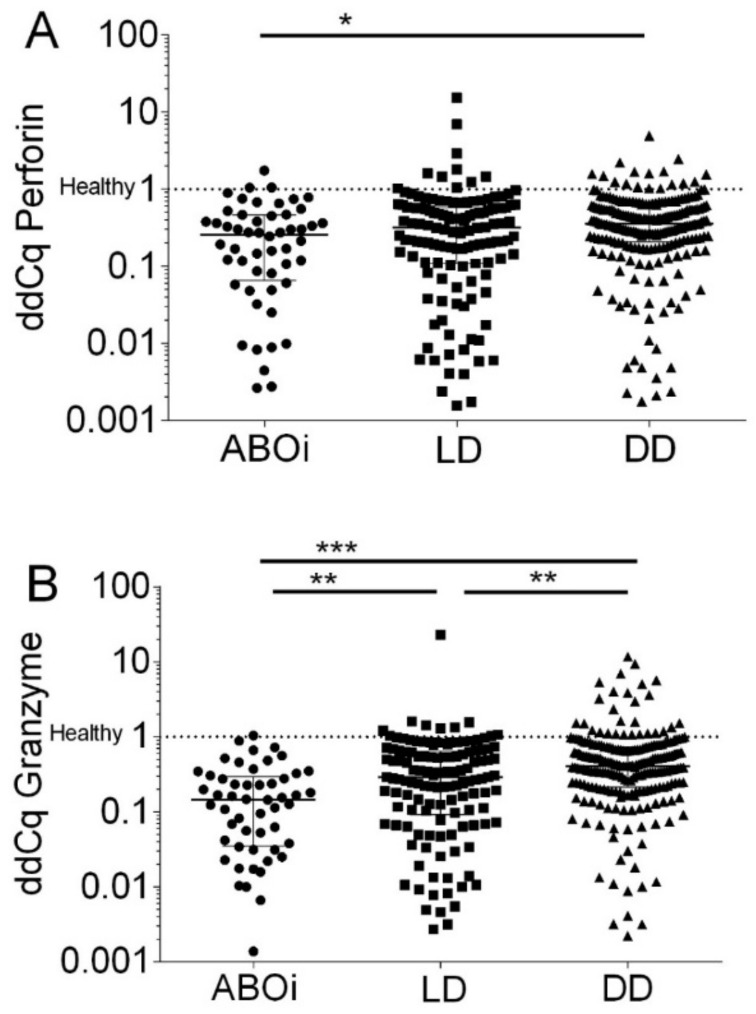
Normalized Perforin transcripts (**A**) and Granzyme B transcripts (**B**) in peripheral blood mononuclear leukocytes from 408 recipients 8 days after incident kidney transplantation. A total of52 recipients had blood group AB0-incompatible living donor allografts (ABOi), 130 recipients had ABO-compatible living donor allografts (LD), and 226 recipients had deceased donor allografts (DD). The Cq values were normalized to 90 healthy control subjects according to the delta–delta Cq (ddCq) method (2^−∆∆Cq^ method) to calculate the relative fold gene expression of samples. Median values and interquartile range (IQR) are indicated. Differences between groups were analyzed with the Kruskal–Wallis test and Dunn’s multiple comparisons test, * *p* < 0.05; ** *p* < 0.01; *** *p* < 0.001).

**Figure 5 biomedicines-12-01258-f005:**
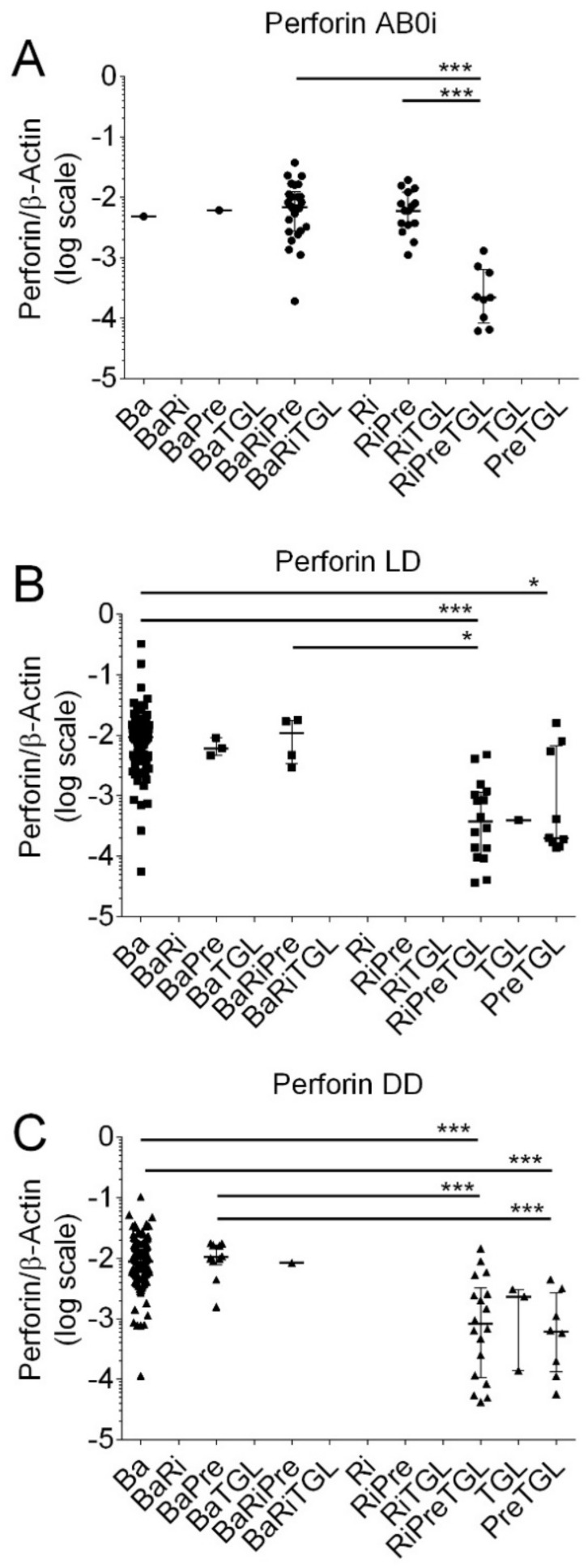
Scatter diagram of Perforin transcripts in peripheral blood mononuclear leukocytes from 408 recipients 8 days after incident kidney transplantation. (**A**) A total of 52 recipients had blood group AB0-incompatible living donor allografts (ABOi), (**B**) 130 recipients had ABO-compatible living donor allografts (LD), and (**C**) 226 recipients had deceased donor allografts (DD). Induction therapy contained combinations of basiliximab (Ba), rituximab (Ri), methylprednisolone (Pre), and thymoglobulin (TGL). Median values and interquartile range (IQR) are indicated. Differences between groups were analyzed with the Kruskal–Wallis test and Dunn’s multiple comparisons test, * *p* < 0.05; *** *p* < 0.001).

**Figure 6 biomedicines-12-01258-f006:**
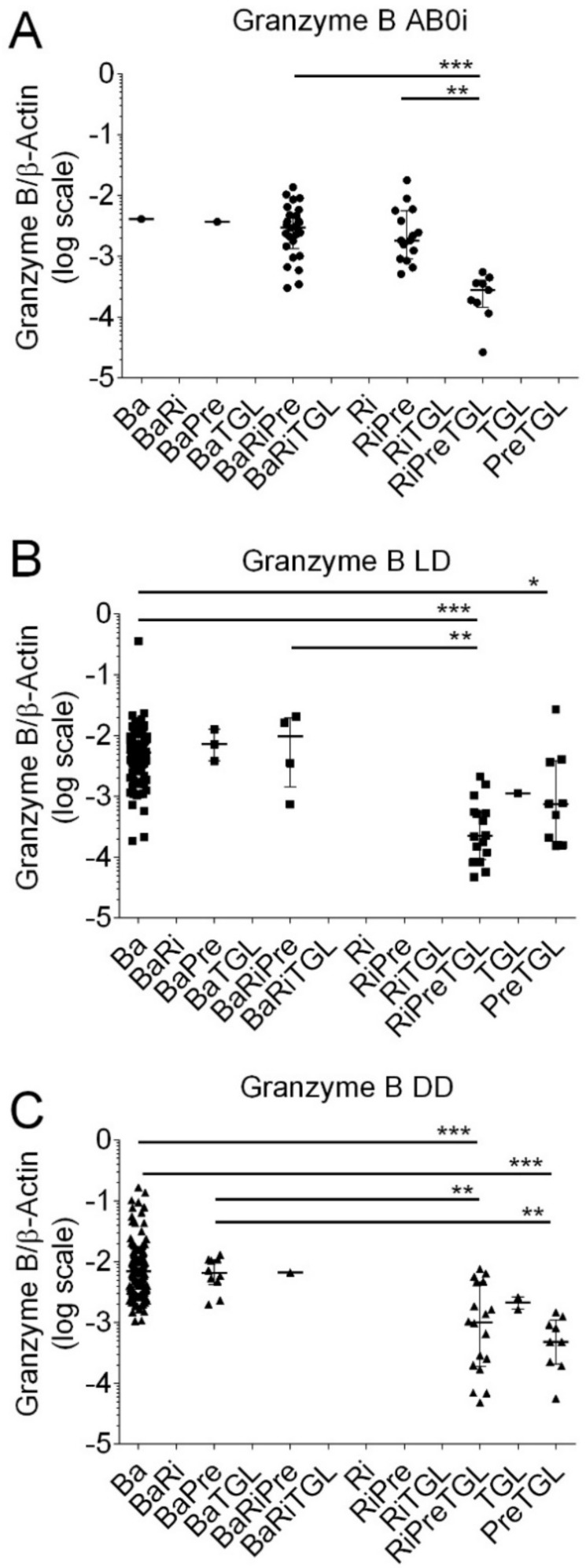
Scatter diagram of Granzyme B transcripts in peripheral blood mononuclear leukocytes from 408 recipients 8 days after incident kidney transplantation. (**A**) A total of 52 recipients had blood group AB0-incompatible living donor allografts (ABOi), (**B**) 130 recipients had ABO-compatible living donor allografts (LD), and (**C**) 226 recipients had deceased donor allografts (DD). Induction therapy contained combinations of basiliximab (Ba), rituximab (Ri), methylprednisolone (Pre), and thymoglobulin (TGL). Median values and interquartile range (IQR) are indicated. Differences between groups were analyzed with the Kruskal–Wallis test and Dunn’s multiple comparisons test, * *p* < 0.05; ** *p* < 0.01; *** *p* < 0.001).

**Figure 7 biomedicines-12-01258-f007:**
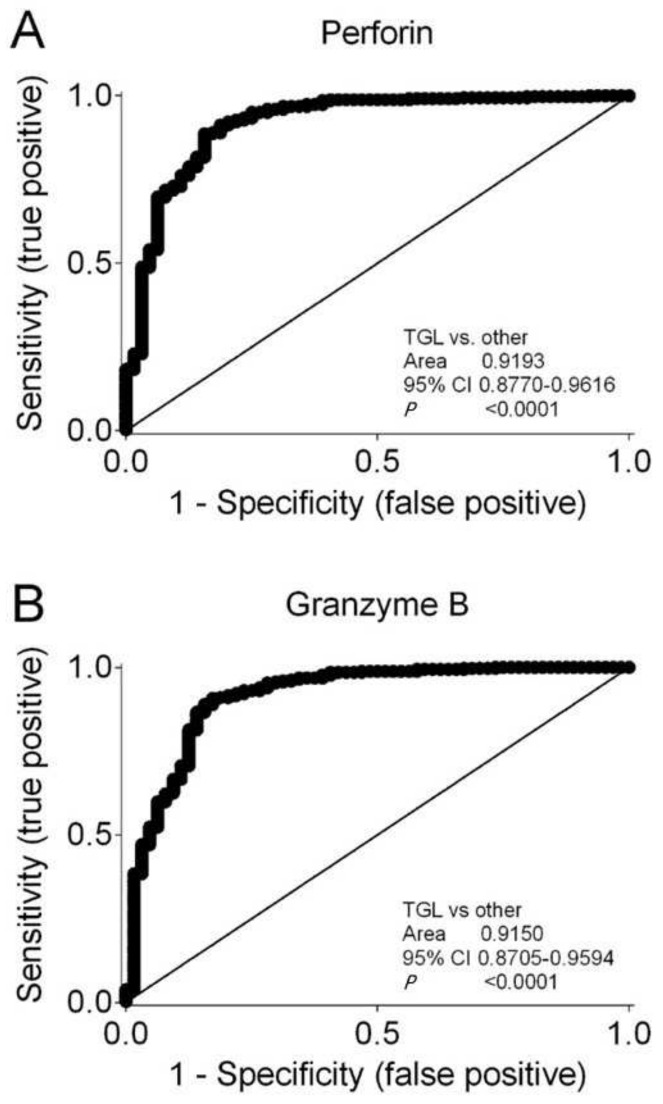
Receiver operating characteristic (ROC) curves showing that (**A**) Perforin and (**B**) Granzyme B characterize immunosuppressive induction therapy containing thymoglobulin compared to all other induction therapies. Area under curves with 95% confidence interval (CI) are shown.

**Table 1 biomedicines-12-01258-t001:** Clinical and laboratory characteristics of 408 kidney transplant recipients.

Characteristic	Total (N = 408)
Recipient gender female, N (%)	136 (33%)
Age of recipient, years	52 (41–62)
Weight, kg	81 (70–93)
Height, cm	175 (167–182)
BMI, kg/m^2^	27 (23–30)
Systolic blood pressure, mm Hg	148 (130–160)
Diastolic blood pressure, mm Hg	85 (75–94)
Cause of end-stage renal disease, N (%)	
Glomerulonephritis	137 (34%)
Diabetic nephropathy	66 (16%)
PCKD	58 (14%)
Hypertensive nephropathy	57 (14%)
Interstitial nephritis/reflux/obstructive	21 (5%)
Other/unknown	69 (17%)
Dialysis vintage, months	12 (3–30)
Hemodialysis N (%)	84 (21%)
Peritoneal dialysis N (%)	222 (54%)
Preemptive transplant N (%)	102 (25%)
Plasma creatinine pretransplant, µmol/L	727 (550–918)
Plasma creatinine first postoperative day, µmol/L	436 (288–609)
Plasma creatinine day 29, µmol/L	142 (113–182)
DGF, N (%)	51 (13%)
Induction therapy, N (%)	
Basiliximab	328 (80%)
Rituximab	89 (22%)
Methylprednisolone	121 (30%)
Thymoglobulin	64 (16%)
ABO-incompatible living donor N (%)	52 (13%)
ABO-compatible living donor N (%)	130 (32%)
Deceased donor N (%)	226 (55%)
Number of HLA mismatches, range	3 (2–4)

Continuous data are presented as median (IQR). Categorical data are presented as numbers (%). BMI, body mass index; PCKD, polycystic kidney disease; DGF, delayed graft function; HLA, human leukocyte antigen. Donor gender and age were available for 255 transplants.

**Table 2 biomedicines-12-01258-t002:** Additional clinical and laboratory characteristics of 408 kidney transplant recipients.

Characteristic	Blood Group AB0-Incompatible Living Donor (N = 52)	Blood Group AB0-Compatible Living Donor (N = 130)	Deceased Donor (N = 226)
Recipient gender female, N (%)	13 (25%)	47 (36%)	76 (34%)
Age of recipient, years	48 (37–59)	45 (35–57)	57 (47–64)
Weight, kg	86 (75–97)	81 (69–94)	80 (69–91)
Height, cm	174 (168–184)	176 (166–183)	174 (167–180)
BMI, kg/m^2^	28 (25–31)	26 (23–30)	27 (23–29)
Systolic blood pressure, mm Hg	148 (134–169)	148 (130–162)	147 (130–160)
Diastolic blood pressure, mm Hg	87 (76–96)	88 (79–96)	83 (74–90)
Cause of end-stage renal disease, N (%)			
Glomerulonephritis	17 (33%)	56 (43%)	64 (28%)
Diabetic nephropathy	7 (13%)	20 (15%)	39 (17%)
PCKD	6 (12%)	12 (9%)	40 (18%)
Hypertensive nephropathy	7 (13%)	15 (12%)	35 (15%)
Interstitial nephritis/reflux/obstructive	4 (8%)	10 (8%)	7 (3%)
Other/unknown	11 (21%)	17 (14%)	41 (18%)
Dialysis vintage, months	7 (0–21)	10 (2–18)	17 (5–36)
Hemodialysis	14 (27%)	29 (22%)	41 (18%)
Peritoneal dialysis	24 (46%)	72 (55%)	126 (56%)
Preemptive transplant	14 (27%)	29 (22%)	59 (26%)
Plasma creatinine pretransplant, µmol/L	721 (584–814)	771 (622–1043)	676 (522–861)
Plasma creatinine first postoperative day, µmol/L	344 (231–445)	353 (253–513)	507 (366–673)
Plasma creatinine day 29, µmol/L	119 (99–150)	132 (113–165)	153 (123–215)
DGF, N (%)	3 (6%)	12 (9%)	36 (16%)
Induction therapy, N (%)			
Basiliximab	28 (54%)	104 (80%)	196 (87%)
Rituximab	50 (96%)	20 (15%)	19 (8%)
Methylprednisolone	51 (98%)	32 (25%)	38 (17%)
Thymoglobulin	9 (17%)	26 (20%)	29 (13%)
Number of HLA mismatches, range	4 (3–5)	3 (2–5)	3 (2–4)

**Table 3 biomedicines-12-01258-t003:** Induction therapies in 408 kidney transplant recipients.

Induction Therapies	Blood Group AB0-Incompatible Living Donor N = 52	Blood Group AB0-Compatible Living Donor N = 130	Deceased Donor N = 226
Basiliximab, N (%)	1 (2%)	97 (75%)	186 (82%)
Basiliximab and prednisolone, N (%)	1 (2%)	3 (2%)	10 (4%)
Basiliximab, rituximab, and prednisolone, N (%)	26 (50%)	4 (3%)	1 (0.5%)
Rituximab and prednisolone, N (%)	15 (29%)	--	--
Rituximab, prednisolone, and thymoglobulin, N (%)	9 (17%)	16 (12%)	18 (8%)
Prednisolone and thymoglobulin, N (%)	--	9 (7%)	9 (4%)
Thymoglobulin, N (%)	--	1 (1%)	2 (1%)

**Table 4 biomedicines-12-01258-t004:** Univariable regression analysis to evaluate clinical and biochemical variables that may predict the ratio of Perforin to Granzyme B transcripts in 226 kidney transplant recipients who received a deceased donor allograft.

Predictor	Univariable B	95% CI	*p*-Value
Recipient age	−0.006	−0.018–0.007	0.377
Recipient gender	−0.187	−0.504–0.130	0–246
Systolic blood pressure	0.0001	−0.008–0.008	0.982
Diastolic blood pressure	0.002	−0.014–0.017	0.841
Relative reduction in creatinine on day 1	0.700	0.136–1.264	0.015
Donor age	0.013	−0.001–0.027	0.071
Donor gender	0.227	−0.191–0.645	0.286
HLA mismatches	0.122	0.012–0.232	0.030

## Data Availability

All data are included in the present manuscript.

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
