# Peer review of "Induction Therapies Determine the Distribution of Perforin and Granzyme B Transcripts in Kidney Transplant Recipients"

_biomedicines, 2024, doi:10.3390/biomedicines12061258_

Round 1

Reviewer 1 Report

Comments and Suggestions for Authors

Martin Tepel et al. reported an interesting work about kidney transplantation. The topic was less frequently discussed, but of a certain importance. The idea was meaningful. Overall, the paper fell within the scope of Biomedicines. A Minor Revision was needed before the final acceptance. Please refer to the following comments:

1.      Please consider to use the manuscript template of Biomedicines to improve the formatting issues.

2.      The abbreviations like ABO should be accompanied with the full name in the Abstract.

3.      The novelty and significance of this work should be mentioned at the end of the Introduction.

4.      The implications for future clinical application should be discussed at the end of the Discussion.

5.      Please add a Conclusion Section.

6.      It would be advisable to redraw Figure 3 as a Pareto chart to showcase the cumulative distribution.

Author Response

Comment Reviewer1

  1. Martin Tepel et al. reported an interesting work about kidney transplantation. The topic was less frequently discussed, but of a certain importance. The idea was meaningful. Overall, the paper fell within the scope of Biomedicines. A Minor Revision was needed before the final acceptance. Please refer to the following comments:
  2. Please consider to use the manuscript template of Biomedicinesto improve the formatting issues.

Answer

For the Revision we used the manuscript template by Biomedicines.

  1. The abbreviations like ABO should be accompanied with the full name in the Abstract.

Answer

In the revised manuscript we added the full text for the abbreviations. For example, we added the expression “blood group ABO”.

  1. The novelty and significance of this work should be mentioned at the end of the Introduction

Answer

At the end of the introduction, we added the sentence, showing the novelty and significance:

We showed for the first time that immunosuppressive induction therapies as well as kidney function were associated with Perforin and Granzyme B transcript levels posttransplant.

  1. The implications for future clinical application should be discussed at the end of the Discussion.

Answer

At the end of the Discussion section we added that we observed that several induction therapies have distinct effects on immune cells. Future studies should determine whether these differences may have an impact on both kidney function as well as complications after kidney transplant, for example, specific infections.

  1. Please add a Conclusion Section.

Answer

We added a conclusion section in the revised manuscript. We conclude clinical parameters and therapies affect Perforin and Granzyme B transcripts posttransplant.

  1. It would be advisable to redraw Figure 3 as a Pareto chart to showcase the cumulative distribution.

Answer

Figure 3 was redrawn showing individual bars and the cumulative distribution.

Reviewer 2 Report

Comments and Suggestions for Authors

The paper is interesting concerning the analysis of granulocytes activity after different induction therapies. The comparison of the methods in this aspect is of interest. 

The introduction is poorly written. It should describe  the advantages and risks related to induction therapies (including risk of infections).

In my opinion the decreased granulocytes activity reflects the efficacy and even more the risk of infections. 

The use of few agent should exclude patients with other groups.

I see no reason for multivariate regression analysis. 

There is no data concerning immunosupression regiments

Please present clinical and laboratory characteristics in table 1 in the way as in table 2 and the flow chart. There is no reason to present the combined data.

It seems reasonable to combine some induction therapies: to Basiliximab based, Rituximab based, ATG based, and mixed (basiliximab with rituximab and rituximab with ATG).  In addition the finding should illustrate perforin and granzyme in this subgroups.

There is no data on infections and rejections during at least first 3 months. (better 12 months). Very useful would be assessment of infection frequency in relation to the expression of perforin and granzyme B.

A large part of the currently presented results should be deleted.

Discussion needs correction in line with new data presentation.

Conclusion is to be created. 

Minor:

Preemptive transplant should not be presented as a type of dialysis therapy.

There is no reason to present relative change in plasma creatinine ratio in the characteristics of patients.

Author Response

Reviewer 2

  1. The paper is interesting concerning the analysis of granulocytes activity after different induction therapies. The comparison of the methods in this aspect is of interest. 

The introduction is poorly written. It should describe the advantages and risks related to induction therapies (including risk of infections). In my opinion the decreased granulocytes activity reflects the efficacy and even more the risk of infections. The use of few agent should exclude patients with other groups.

Answer

In the introduction we added that different induction therapies may show different risk for infections.

There are several induction therapies in kidney transplant recipients, including basiliximab, rituximab, prednisolone, and thymoglobulin. Induction therapies in kidney transplant recipients are chosen in the basis of clinical experiences and guidelines. It is suspected that different induction therapies may show different side effects including infections. Since Perforin and Granzyme B are major players in the defense against pathogens induction therapies may affect infections by interfering with Perforin and Granzyme B. Currently, it is unknown whether different immunosuppressive induction therapies, including basiliximab, rituximab, prednisolone, and thymoglobulin, may cause separate effects on Perforin and Granzyme B transcripts in peripheral blood mononuclear cells. The objective of the present study was to compare the effects of immunosuppressive induction therapies on Perforin and Granzyme B transcripts posttransplant in mononuclear cells from kidney transplant recipients.

  1. I see no reason for multivariate regression analysis. 

Answer

As suggested by the reviewer we omitted multivariable regression in the manuscript.

  1. There is no data concerning immunosuppression regiments

Answer

To indicate the immunosuppressive therapies, we included that

328 kidney transplant recipients (80%) were treated with basiliximab (Ba), 89 recipients (22%) rituximab (Ri), 121 recipients (30%) methylprednisolone (Pre), and 64 recipients (16%) thymoglobulin (TGL), respectively. The total number exceeds 100% be-cause recipients obtained combinations of more than one immunosuppressive agent. In 52 kidney transplant recipients with ABOi 1 was treated with Ba (2%), 1 BaPre (2%), 26 BaRiPre (50%), 15 RiPre (29%), and 9 RiPreTGL (17%), respectively. In 130 kidney transplant recipients with LD 97 were treated with Ba (75%), 3 BaPre (2%), 4 BaRiPre (3%), 16 RiPreTGL (12%), 1 TGL (1%), and 9 PreTGL (7%), respectively. In 226 kidney transplant recipients with DD 186 were treated with Ba (82%), 10 BaPre (4%), 1 BaRiRe (0%), 18 RiPreTGL (8%), 2 TGL (1%), and 9 PreTGL (4%), respectively.

  1. Please present clinical and laboratory characteristics in table 1 in the way as in table 2 and the flow chart. There is no reason to present the combined data.

Answer

The new Table 2 shows clinical and laboratory characteristics according to blood group, i.e., AB0-incompatible living donor; AB0-compatible living donor; Deceased donor.

  1. It seems reasonable to combine some induction therapies: to Basiliximab based, Rituximab based, ATG based, and mixed (basiliximab with rituximab and rituximab with ATG).  In addition the finding should illustrate perforin and granzyme in this subgroups.

There is no data on infections and rejections during at least first 3 months. (better 12 months).

The actual distribution of immunosuppressive substances is already depicted in Figures 5 and 6. The effects of induction therapies on Perforin transcripts are depicted in Figure 5. The effects on Granzyme B transcripts are depicted in Figure 6.

The reviewer suggested to “combine some induction therapies”. We analyzed Perforin and Granzyme B in two groups, i.e., one group containing thymoglobulin and another group with all other induction therapies. The results are shown in Figure 7.

Kidney transplant recipients who were treated with thymoglobulin had significantly lower Perforin levels compared to all other induction therapies (median, 0.00046; IQR, 0.00014 to 000187; N=64; vs. median, 0.00865; IQR, 0.00497 to 0.01355; N=344; P<0.001 by Mann-Whitney test). Kidney transplant recipients who were treated with thymoglobulin also had significantly lower Granzyme B levels compared to all other induction therapies (median, 0.00048; IQR, 0.00017 to 000136; N=64; vs. median, 0.00590; IQR, 0.00320 to 0.01045; N=344; P<0.001 by Mann-Whitney test).

  1. Very useful would be assessment of infection frequency in relation to the expression of perforin and granzyme B.

Answer

Data on specific infections were not available.

  1. A large part of the currently presented results should be deleted. Discussion needs correction in line with new data presentation.

Answer

Results are main important parts of the manuscript for every reader. We described the findings accurately to underline the importance of Perforin and Granzyme B in kidney transplant recipients.

  1. Conclusion is to be created.

Answer

A conclusion was added. We conclude clinical parameters and therapies affect Perforin and Granzyme B transcripts posttransplant.

Minor:

  1. Preemptive transplant should not be presented as a type of dialysis therapy.

Answer

Preemptive dialysis is given in a separate row.

  1. There is no reason to present relative change in plasma creatinine ratio in the characteristics of patients.

Answer

Relative change of creatinine is omitted in the presentation of patient characteristics.

Reviewer 3 Report

Comments and Suggestions for Authors

Comments on the manuscript titled: Induction therapies determine the distribution of Perforin and Granzyme B transcripts in kidney transplant recipients by Dino Pipic , Marianne Rasmussen, Qais W. Saleh and Martin Tepel

It is an interesting manuscript that continues the previous Authors' studies.

The main findings of the study regarding the levels of Perforin transcripts in kidney transplant recipients show that the distribution of Perforin transcripts in peripheral blood mononuclear leukocytes varies depending on the type of induction therapy received by the recipients The study found that recipients with ABO-incompatible living donor allografts (ABOi) had different levels of Granzyme B transcripts compared to recipients with ABO compatible living donor allografts (LD) and deceased donor allografts (DD). It was found that the type of donor compatibility may influence the expression of Granzyme B transcripts in kidney transplant recipients. The implications of these findings suggest that the choice of induction therapy and the type of donor compatibility in kidney transplantation may impact the expression of key immune response genes such as Perforin and Granzyme B

Understanding these differences can help to choose immunosuppressive therapies to optimize outcomes for kidney transplant recipients.

Remarks:

Introduction

The introduction needs more explanation about why immunosuppressive therapy is given to patients and how it depends on blood types. More details are required concerning the patient's consequences of applying certain immunosuppressive therapies.

Ref [13] on page 3 is misleading as it looks as the Authors studies while it is a different group. Please re-write for clarity.

Materials and Methods

Please explain what routine laboratory data means in part Study design

Please give more details concerning experiments

If centrifugation is executed please state a centrifuge name, producer, rotor, rpm, and that g.

Figure 1 is not necessary as all the details are placed within the text.

Spectrophotometric measurements-please give details on baseline measurement/ correction

Nuclease-free water - state producer and Country

Statistical analyses

Explain in detail on what condition Mann -Whitney test and Kruskall-Wallis tests were executed. What was the reason on choosing such an analysis?

The same with Fishers exact and chi-squared.

Explain in details why multivariable models were executed and what is the difference if data is analysed with univariate analyses.

If Ur manuscript is accepted, before it is published with the detailed answers to the remarks please provide the following changes:

Please organize Ur manuscript according to the Instructions for Authors

Use a Template provided in Word or LaTeX format

Note ABO and AB0 throughout the text – correct

Please unify the manuscript's language so that the whole is from the use of the passive or personal side. Do not mix the two ways in the same paragraphs or even sentences.

Research manuscripts should comprise:

-Front matter: Title, Author list, Affiliations, Abstract, Keywords.

- Research manuscript sections: Introduction, Materials and Methods, Results, Discussion, Conclusions (optional), Patents.

- Back matter: Supplementary Materials, Acknowledgments, Author Contributions, Conflicts of Interest, References.

Other remarks: J.B. Winsløwsvej 21.3 is this the right street number?

Comments on the Quality of English Language

Language is fine

Author Response

Reviewer 3

Comments on the manuscript titled: Induction therapies determine the distribution of Perforin and Granzyme B transcripts in kidney transplant recipients by Dino Pipic , Marianne Rasmussen, Qais W. Saleh and Martin Tepel

It is an interesting manuscript that continues the previous Authors' studies.

The main findings of the study regarding the levels of Perforin transcripts in kidney transplant recipients show that the distribution of Perforin transcripts in peripheral blood mononuclear leukocytes varies depending on the type of induction therapy received by the recipients The study found that recipients with ABO-incompatible living donor allografts (ABOi) had different levels of Granzyme B transcripts compared to recipients with ABO compatible living donor allografts (LD) and deceased donor allografts (DD). It was found that the type of donor compatibility may influence the expression of Granzyme B transcripts in kidney transplant recipients. The implications of these findings suggest that the choice of induction therapy and the type of donor compatibility in kidney transplantation may impact the expression of key immune response genes such as Perforin and Granzyme B

Understanding these differences can help to choose immunosuppressive therapies to optimize outcomes for kidney transplant recipients.

Remarks:

  1. Introduction

The introduction needs more explanation about why immunosuppressive therapy is given to patients and how it depends on blood types. More details are required concerning the patient's consequences of applying certain immunosuppressive therapies.

Answer

There are several induction therapies in kidney transplant recipients, including basiliximab, rituximab, prednisolone, and thymoglobulin. Induction therapies in kidney transplant recipients are chosen in the basis of clinical experiences and guidelines. It is suspected that different induction therapies may show different side effects including infections. Since Perforin and Granzyme B are major players in the defense against pathogens induction therapies may affect infections by interfering with Perforin and Granzyme B. Currently, it is unknown whether different immunosuppressive induction therapies, including basiliximab, rituximab, prednisolone, and thymoglobulin, may cause separate effects on Perforin and Granzyme B transcripts in peripheral blood mononuclear cells. The objective of the present study was to compare the effects of immunosuppressive induction therapies on Perforin and Granzyme B transcripts posttransplant in mononuclear cells from kidney transplant recipients.

  1. Ref [13] on page 3 is misleading as it looks as the Authors studies while it is a different group. Please re-write for clarity.

Answer

For clarity this part was separated into 2 sentences.

We investigated the effects of different induction therapies. According to the literature the relative reduction of creatinine was used as an outcome measure [13].

  1. Materials and Methods

Please explain what “routine laboratory data” means in part Study design

Please give more details concerning experiments

If centrifugation is executed please state a centrifuge name, producer, rotor, rpm, and that g.

Answer

The term “routine” was omitted.

The quantitative reverse transcriptase real-time polymerase chain reaction (qRT-PCR) is given exactly with primers, cycling conditions, analyzes and calculations.

  1. Figure 1 is not necessary as all the details are placed within the text.

Spectrophotometric measurements-please give details on baseline measurement/ correction

Nuclease-free water - state producer and Country

Answer

Figure 1 shows the common flow chart of incident kidney transplant recipients, included patients, excluded patients, and the different kidney transplant groups at a glance.

  1. Statistical analyses

Explain in detail on what condition Mann -Whitney test and Kruskall-Wallis tests were executed. What was the reason on choosing such an analysis?

The same with Fisher’s exact and chi-squared.

Explain in details why multivariable models were executed and what is the difference if data is analysed with univariate analyses.

Answer

We added that we used non-parametric test to compare two and more than two groups, respectively. Fisher’s exact test was used for contingency tables. According to reviewer’s comments (see above) multivariate analyses were omitted.

Round 2

Reviewer 2 Report

Comments and Suggestions for Authors

The paper was improved. In my opinion, the authors should add limitations concerning a lack of infection analysis and the need to address such data in further research.

Author Response

Answer

We added a limitation that it lacks an infection analysis and such an analysis should be performed in future investigations.

Reviewer 3 Report

Comments and Suggestions for Authors

Thank U very much. All my questions have been answered

Comments on the Quality of English Language

Language fine

Author Response

We do thank the reviewer for the insightful comments.